# Research on Using K-Means Clustering to Explore High-Risk Products with Ethylene Oxide Residues and Their Manufacturers in Taiwan

**DOI:** 10.3390/foods13162510

**Published:** 2024-08-11

**Authors:** Li-Ya Wu, Fang-Ming Liu, Wen-Chou Lin, Jing-Ting Qiu, Hsu-Yang Lin, King-Fu Lin

**Affiliations:** Food and Drug Administration, Ministry of Health and Welfare, Taipei 115209, Taiwan; lywu@fda.gov.tw (L.-Y.W.); wenjou@fda.gov.tw (W.-C.L.); maday852@fda.gov.tw (J.-T.Q.); michael@fda.giv.tw (H.-Y.L.); rich0822@fda.gov.tw (K.-F.L.)

**Keywords:** K-means clustering, food safety, ethylene oxide, unsupervised learning

## Abstract

Considering the frequency of ethylene oxide (EtO) residues found in food, the health effects of EtO have become a concern. Between 2022 and 2023, 489 products were inspected using the purposive sampling method in Taiwan, and nine unqualified products were found to have been imported; subsequently, border control measures were enhanced. To ensure the safety of all imported foods, the current study used the K-means clustering method for identifying EtO residues in food. Data on finished products and raw materials with EtO residues from international public opinion bulletins were collected for analysis. After matching them with the Taiwan Food Cloud, 90 high-risk food items with EtO residues and 1388 manufacturers were screened. The Taiwan Food and Drug Administration set up border controls and grouped the manufacturers using K-means clustering in the unsupervised learning algorithm. For this study, 37 manufacturers with priority inspections and 52 high-risk finished products and raw materials with residual EtO were selected for inspection. While EtO was not detected, the study concluded the following: 1. Using international food safety alerts to strengthen border control can effectively ensure domestic food safety; 2. K-means clustering can validate the risk-based purposive sampling results to ensure food safety and reduce costs.

## 1. Introduction

In August 2022, the European Commission revised the law to regulate the residual amount of ethylene oxide (EtO) in food to within 0.1 mg/kg, thereby circumventing the previous law enforcement difficulties caused by the unclear source of EtO. Since then, the EU Rapid Alert System for Food and Feed (RASFF) has successively reported many foods containing EtO residues, such as sesame, ice cream, flour products, bread, biscuits, spice plants, curry, empty capsules for food use, and other products [1]. In other countries, including Japan and South Korea [2,3], the maximum residue level must be below 0.01 ppm in food products, while the corresponding values in Canada are 7 ppm and 0.1 ppm for dried vegetables and other foods, respectively [4]. In order to maintain food safety and prevent products that may contain EtO residues from entering the country, the Taiwan Food and Drug Administration (TFDA) announced an inspection of products that could contain the pesticide residue on 14 March 2022 and has not approved the use of EtO as a pesticide. In addition, the Taiwan Use Scope, Limits, and Specifications of Food Additives policy does not allow the use of EtO gas as a food additive; so, it cannot be detected in food [5]. EtO is a colorless flammable gas that is soluble in water and other organic solvents and reacts readily with chlorine to produce 2-chloroethanol (2-CE). It is mainly used in the production of ethylene glycol (as an antifreeze), alongside being used for the sterilization of medical equipment, pharmaceuticals, and cosmetics. It blocks cell metabolism and replication by alkylating proteins and nucleic acids, which in turn leads to the death of microorganisms, hence achieving bactericidal effects [6].

The International Agency for Research on Cancer (IARC) has classified EtO as an IARC Group 1 carcinogen, which can enter the human body through inhalation, ingestion, and skin contact. The Department of Labor Occupational Safety and Health Administration (OSHA) has stated that exposure to EtO can cause symptoms such as dizziness, headache, nausea, vomiting, coughing, convulsions, blistering, and even difficulty breathing and blurred vision, while occupational exposure to EtO is associated with lymphoma, leukemia, and breast cancer [7,8,9]. Although the United States and Canada currently allow EtO to be used as a fumigant for crops such as sesame, spices, mint, and dried vegetables, Taiwan, European Union, Japan, and Australia have not approved its use as a pesticide.

Currently, the risk of residual EtO in food is an international concern. The food cloud database was established by the TFDA in 2015. The main purpose of its establishment was to assist health authorities in quickly obtaining product flow information from the database when a problematic product occurs to prevent it from entering the market. The food cloud includes the registration data of suppliers, upstream and downstream flow data of products in the market, inspection data (or import data) of border products, product inspection data, raw product materials, and ingredient data.

To determine whether high-risk raw materials or food products notified by the European Union and United States have the same situation in the domestic market, the current study aimed to investigate whether Taiwan has similar or identical risks of EtO residues in food and to enhance active inspection against risky products or manufacturers through big data risk analysis. This evidence may help prevent problematic foods from entering the market and affecting overall consumer health.

According to the Taiwan Import Food Information System (IFIS) data, from 1 March 2022 to 28 February 2023, a total of 1623 tests for EtO residues were conducted on foods at Taiwan’s borders imported from 46 countries. Of these, 44 batches were found to have EtO residues exceeding the permissible limits, resulting in a failure rate of 2.71%. Food imported from eight countries contained EtO residues. The failure rates in these countries were 1.45% in Japan, 2.03% in South Korea, 5.03% in Vietnam, 12.16% in Indonesia, 1.57% in France, 3.03% in Switzerland, 11.76% in the Philippines, and 33.33% in Brazil. The largest number of unqualified batches was 18 in Indonesia, and the highest unqualified rate was 33.33% in Brazil [10].

Before the implementation of this study, the TFDA had successfully intercepted imported batches of sesame, instant noodles, cheese, ice cream, and spicy seasoning with EtO residues at the border through the artificial intelligence assistance of the EL V2 risk-prediction model and requested them to be returned or destroyed [11]. For the same problematic products (including different batch numbers) that have flowed into the country and are still within the validity period, the flow was immediately traced, and random inspections were conducted.

The most common sampling methods used for food safety purposes included simple random sampling, purposive sampling based on risk, or model-assisted sampling [12,13]. The safety control measures taken by the TFDA for post-market foods included random on-site inspections of products and raw materials. The random inspection method is “risk-based purposive sampling” using the experience of senior inspectors. Statistical analysis of spot inspection data of Taiwan’s food from 2022 to 2023 showed that nine out of four-hundred and eighty-nine products contained EtO residues in the post-market stage (Table A1). The nine unqualified cases included one sesame case, four spice cases (including spice seasonings and curry powder), two cheese cases, and two instant noodle cases (Table 1). These problematic cases involved imported products that must be recalled, destroyed, or withdrawn from the market to ensure the safety of public consumption.

## 2. Materials and Methods

### 2.1. Choice of Analytical Method for Sampling

If we can identify the risky food items with EtO residues that have been detected internationally, we can use food cloud big data to obtain information on the manufacturers and sellers who produce or sell the same raw food materials and products in Taiwan. Another consideration would be the type of risk analysis method that is used to plan the “Priority Inspection List”.

K-means clustering is an unsupervised learning algorithm developed by MacQueen in 1967 and is one of the most widely used clustering analysis methods [14]. It is a major tool in the field of data mining. Unsupervised learning is the most commonly used prediction method in the absence of learning objects. Since there is no standard answer in the process of finding rules through the training data, the algorithm must find a set of grouping rules from the data group by itself. The advantage of unsupervised learning is that data need not be labeled in advance, and only data features input into the algorithm are required when training the model. It can then automatically determine the rules of classification and divide the data into K-groups.

The K-means clustering algorithm is widely used in many fields, such as historical retail transaction data clustering [15], abnormal cell image segmentation detection [16], and commercial building temperature clustering [17]. Mohsen et al. used 143 Sentinel-2 images to analyze the dynamics of surface-suspended sediments using a K-means unsupervised classification algorithm [18]. Miolla et al. investigated the consumers’ acceptance of bread enriched with oenological by-products by applying K-means clustering, thereby providing insights into the characterization of consumer groups based on their specific features and declared attitudes [19]. To improve the quality assessment of wheat during storage, Liu collected and analyzed the data from more than 20 regions, including information on storage environmental parameters and changes in wheat pesticide residue concentrations. Based on these factors, a model was developed to predict the changes in wheat pesticide residue concentrations during storage. A comprehensive wheat quality assessment index was set for the predicted and true values of pesticide residue concentrations, and it was then combined with the K-means algorithm to assess the quality of wheat during storage. The results of this study demonstrated that the model achieved optimal prediction results and the smallest error values [20]. Ferreira-Paiva et al. proposed an urban clustering approach—namely, K-means clustering—to identify the most critical sectors related to greenhouse gas emissions, based on the percentage contribution of agriculture, land-use change, energy, and waste sectors to total emissions [21]. These studies were based on the selected characteristic factors, and a cluster analysis was performed on the target groups to classify and group them. This approach is commonly known as “birds of a feather flock together”. It can help judge and analyze the characteristics and properties of each group according to the characteristic factors. Therefore, based on the previous studies, the current study used the K-means cluster analysis method to identify risk vendors. First, we must determine the characteristic factors of risk manufacturers (the premise of this research is that data on risk characteristic factors must exist). We believe that the K-means cluster analysis method would help obtain a list of prioritized manufacturers for inspection.

### 2.2. The Principle of K-Means Clustering

Cluster analysis is used in data mining and machine learning to classify similar objects into clusters. K-means clustering is a widely used cluster analysis method that divides a set of objects into K-clusters. The sum of squared distances between the objects and their assigned clusters is then minimized. K-means clustering is a method of grouping data by calculating their similarity. This similarity is based on the Euclidean distance formula in n-dimensional space to calculate the distance between each data point and the nearest cluster center [22]. A K-means clustering algorithm was selected for this study since its theory is comprehensible. The steps of operation of this algorithm are as follows: First, a dataset is divided into K-clusters, the cluster center (CC) is randomly selected, and the Euclidean distance (ED) to the cluster center is calculated for each data point. Next, each data point is assigned to the nearest cluster center to form a cluster. Subsequently, the new cluster centers are recalculated and updated [23]. The above steps are repeated until none of the cluster centers change. From the above steps, the entire process involves the classification of similar data into the same cluster and dissimilar data into different clusters.

First of all, a set of n d-dimensional data, $x_{i}\in R^{d},i=1,2,3,..,n$, and artificially set K (must be ≤n) clusters {**S**1, **S**2,…, **S**k} are assumed; then, K-means clustering is expected to minimize the distance within the cluster and make the sum of squared errors of cluster centers as small as possible. The $\mu_{c}$ is the cluster center, and ‖x − y‖ calculates the ED. The K-means clustering algorithm can be defined by the following formulas (Formulas (1)–(4)):Initially, K cluster centers are set randomly
(1)$$\mu_{c}^{\left(0\right)}\in R^{d},c=1,2,\ldots ,K$$The distance between each sample and each cluster center is calculated, with (t) being the t-th operation. Then, each sample is assigned to the cluster with the shortest distance.
(2)$$S_{c}^{\left(t\right)}=\left\{x_{i}:\left\|x_{i}-\mu_{c}^{\left(t\right)}\right\|\leq \left\|x_{i}-\mu_{c*}^{\left(t\right)}\right\|,\forall i=1,\ldots ,n\right\}$$CC is updated (n_c_ data are in cluster c)
(3)$$\mu_{c}^{\left(t+1\right)}=\frac{sum(s_{c}^{\left(t\right)})}{n_{c}}=\sum_{i=1}^{n_{c}}x_{i}\left|\begin{array}{cc} x_{i\in }s_{c}^{\left(t\right)} & \end{array}\right.$$Steps 2–3 are repeated until the CC does not change—that is,
(4)$$S_{C}^{\left(t+1\right)}=S_{C}^{\left(t\right)},\;\forall c=1,\ldots ,K$$

The greatest challenge in unsupervised learning is to determine the K-value (number of clusters) [24]. This study used the elbow method to determine the K values. The elbow method must be executed before using the K-means clustering algorithm [25]. In the same group of data, continuous increments by 1 are made from K = 2. Initially, the curve decreases rapidly. When K reaches a certain value, the curve begins to decline slowly. The first K value after slowdown is the optimal value [26] (Figure 1). This concept involves calculating the sum of squared errors (SSE) for each value of K to estimate the best elbow point [27]. The formula is as follows (Formula (5)):(5)$$\mathrm{SSE}=\sum_{i-1}^{K}\sum_{p\in Ci}{\left|p-mi\right|}^{2}$$

K: total number of clusters;

Ci: represents one of the clusters;

mi: central point of the cluster.

### 2.3. Data Sources and Analytical Tools

In order to control the risk of EtO residues in food, the current study used the collected international food safety alerts and changed the previous purposive sampling to an unsupervised learning method to conduct “food safety risk prediction”. Improvements in post-market sampling methods further optimized the selection of riskier samples. The research methodology is described in the following sections in detail.

The data used in this study were mainly obtained from real-time open information in the food cloud. The collected content included the names of raw materials or finished products, date of notification, country, and manufacturers. These data were collected from the Consumer Agency of Japan, Rappel Conso of France, Food Safety Authority of Ireland (FSAI), EU RASFF, US Food and Drug Administration (FDA), USDA’s Food Safety and Inspection Service (FSIS), Food Standards Australia New Zealand (FSANZ), British Food Standards Agency (FSA), Canadian Food Inspection Agency (CFIA), Hong Kong Food and Environmental Hygiene Department Food Safety Center, Singapore Food Agency (SFA), and other international sources. Further, the data sources included supplier registration, inspection, testing, and product flow information in the food cloud established by the Food and Drug Administration of the Taiwan Ministry of Health and Welfare. The food cloud used the five major systems of TFDA as its core, namely, the Registration Platform of Food Businesses System (RPFBS), Food Traceability Management System (FTMS), Inspection Management System (IMS), Product Management Decision System (PMDS), and IFIS (Table 2). In addition, it comprised cross-agency data communication, including financial and tax electronic invoices, customs electronic gate verification data, national business tax registration data, industrial and commercial registration data, indicated chemical substance flow data, domestic industrial oil flow data, imported industrial flow data, waste oil flow data, toxic chemical substance flow data, feed oil flow data, campus food ingredient logins, and inspection data.

Imported foods and food-related products have to apply for import inspection through the IFIS from the TFDA, and only compliant products are permitted to enter the domestic market. The relevant business data must be registered with the RPFBS, national business tax registration data, and business registration data. The flow of information generated by domestic and imported products entering the market from the border should be recorded in the IFIS and FTMS, as well as in electronic invoices and electronic gate goods import and export verification records. All products sampled and inspected by the government were recorded in the PMDS, IFIS, and IMS. A company’s product information can also be obtained through the RPFBS and FTMS [28] (The full name of the abbreviation can be found in Table A2).

The analytical tools used in this study were R.4.0.0, SPSS 25.0, and Microsoft Excel 2010.

### 2.4. Selection of Key Risk Factors

To date, all problematic food products or raw materials with EtO residues found in Taiwan have been imported. In the absence of unqualified cases, we selected products or raw materials that had been reported to be contaminated with EtO abroad as a reference for our sampling project in Taiwan. The collection period for the above information was from 1 January 2021 to 31 March 2023. After integrating the problematic product information reported internationally in this study, 913 problematic products and raw material notifications with EtO residues were identified. Among them, we analyzed 90 clearly identifiable finished products (or categories) and raw materials that were screened out and were within the scope of this study.

The names of the 90 products and raw materials with EtO residues in Table 3 were matched with those of the products and raw materials in the FTMS. A list of risky product manufacturers (or suppliers) was obtained using the aforementioned steps. To plan the “Priority inspection list” for risky product manufacturers, variables associated with the manufacturers were selected to design the key risk factors. The design concept of these factors was that there must be “existing data”, so that the factors can be selected from the food cloud database. Five key risk factors determined by consensus among the experts in this study included “Capital amount (C)”, “Number of occupations (N)”, “Manufacturers that have detected unqualified finished products or raw materials with EtO residues during border inspections (Bb)”, “Quantity of products or raw materials that may be involved in EtO risk (Q)”, and “Manufacturers whose finished products or raw materials have been detected to contain ethylene oxide residues during the post-marketing period (Bp)”.

### 2.5. Research Methodology

To understand whether there are risky finished products or raw materials with EtO residues in Taiwan’s consumer market, the current study used preliminary research references to identify high-risk products and related food manufacturers to protect people from the risk of EtO. The steps of this study are detailed below and illustrated in Figure 2.

Step 1. The research topic was determined. To conduct a preliminary study on whether there are potentially risky finished products or raw materials containing EtO residues in Taiwan.

Step 2. The names of risky finished products or raw materials (RFPRMs) with EtO residues were collected and sorted.

Step 3. RFPRM data were matched with the FTMS data in the food cloud database to obtain a list of domestic manufacturers (or suppliers; MS) who produce or sell the same finished products or raw materials.

Step 4. The key risk factors required for the research were analyzed, matched, and organized. This step included the following sub-steps:MS was used to match the RPFBS data in the food cloud database to obtain key risk factor 1—the Capital Amount (C) of MS. The reason this study used the C-factor is related to the manufacturer’s production volume, product circulation scope, and market share. These products are easily accessible to consumers and, therefore, closely related to food safety.MS was used to match the RPFBS data in the food cloud database to obtain the key risk factor 2—the number of occupations (N) of MS. There are five categories of N factor: import industry, manufacturing industry, logistics industry, catering industry, and sales industry. More occupations mean that products can reach consumers through more extensive circulation channels or scopes. It is closely related to food safety.MS was used to match the IFIS data in the food cloud database to obtain the key risk factor 3—“list of suppliers whose unqualified finished products or raw materials with EtO residues were detected during border inspection (referred to as Bb)”.Calculate the number of finished products and raw materials that may have EtO residues to determine key risk factor 4—“quantity of finished products or raw materials that may involve EtO risk (referred to as Q)”.MS was used to match the PMDS data in the food cloud database to obtain key risk factor 5—“Manufacturers whose finished products or raw materials with residues have been detected in Taiwan’s market (the factor is referred to as Bp)”.

The quantitative interpretation table of the aforementioned five factors is provided in Table 4.

Step 5. K-means clustering was used to cluster MS. Before that, the K-value was determined by asking “How many clusters must be determined first?”. This study used a simple and easy-to-understand elbow method to determine the K-value. First, from K = 2, the value started to gradually increase by 1; then, the SSE was calculated (Table 5) and a graph of the K-value and SSE was created (Figure 3). After K = 4, the curve began to decline slowly, and after K = 5, there was only an extremely small change in the curve. After observing the results, we chose K = 5 for the K-means clustering.

Step 6. The risk scores of each cluster were calculated, and the cluster with the highest cluster center (HCG) was selected as the initial target range for this study.

Step 7. The cluster with the highest risk score for the cluster center was selected, and the distance of each point in the cluster (each point represents a manufacturer or supplier) from the cluster center was calculated. When the distance between a point and the cluster center was small, the similarity between the point and the cluster center was high. In this study, points with shorter distances were prioritized for inspection.

Step 8. The inspection list was organized. After sorting according to Step 7, manufacturers whose target products or raw materials had sales records for the past 3 months in the FTMS were selected.

Step 9. A priority inspection list was obtained, and inspections were initiated. A list of priority inspection manufacturers was used to match the RFPRM produced or sold to facilitate the selection of samples during the inspection.

Step 10. Sampling was performed from February to June 2024, and the inspection results were evaluated.

## 3. Results

### 3.1. High-Risk Manufacturers Obtained after Clustering

To find the target of priority inspection under a limited inspection workforce and resources, this study used the theoretical method of “birds of a feather flock together” in K-means clustering to discover the potential risky product manufacturers, thereby enabling priority inspection. After sorting the international alert notification information from 2020 in this study, there were a total of 913 notifications on problematic finished products and raw materials with EtO residues. After further data screening and cleaning, 90 finished products, categories, and raw material names were identified. These were then matched with the names in the food cloud database to obtain detailed information on 1388 domestic risky product manufacturers (MS).

In this study, the value of K was set to 5 to perform the calculation of K-means clustering for the aforementioned 1388 manufacturers (MS). In the clustering process, waiting until the cluster center is no longer moving is necessary before it can be stopped. This process went through four calculations iteratively (Table 6). The minimum distance between the final cluster centers was 2.731 (clusters 4 and 5), and the longest distance was 8.162 (clusters 4 and 1), as shown in Table 7. This result indicated that the risk profiles of clusters 4 and 5 were more similar than that of cluster 1.

Further analysis of variance was performed on the five key risk factors, and the results showed that the C, N, Bb, and Q factors were suitable for inclusion in this study owing to the significant differences (*** *p* < 0.001) and independence among them. However, BPI was not suitable for inclusion in this study as a key risk factor at this stage, probably because no domestic manufacturer was found to involve EtO residue risk in their finished products or raw materials (Table 8). Based on the above, the four risk factors C, N, Bb, and Q were mainly used in this study for clustering.

The results showed that cluster 4 was the riskiest group of manufacturers. Among the five clusters established, the cluster center (CC) risk score of cluster 4 was 26, which was the highest among all (Table 9), and 212 manufacturers in total were included in the cluster. In this study, the four factors of C, N, Bb, and Q were first quantified, standardized, and unweighted; then, the scores of these four factors were summed up to obtain the risk score. Since K-means clustering did not use Bp in the cluster calculation process, it had a risk score of zero.

### 3.2. The Risk-Ranked List of Manufacturers Obtained by Euclidean Distance

In cluster analysis, each point represents a vendor, and the cluster center (CC) is the most representative virtual core of the cluster. In this study, the ED was calculated between each point in cluster 4 and the cluster center (CC). The smaller the ED, the higher the similarity between each vendor and the CC (the core with the highest risk). In this study, the conditions for selecting the preferred manufacturers for inspection were as follows: 1. Manufacturers with a relatively short ED; 2. More than two items manufactured or sold by the manufacturer must be finished products or raw materials that belong to international risk notification (Table 2); 3. The items manufactured or sold by the manufacturer were evaluated by experts as items frequently consumed by the Taiwanese and were also notified internationally as RFPRMs. Finally, 52 samples were carefully selected for priority sampling in this study, including sesame and its products, oats, vegetable sauce, ketchup, pepper, berry products, baked goods or ingredients, xanthan gum, sodium carboxymethyl cellulose, buckwheat, stevia leaves, coffee beans, chili (dry powder products), chocolate, carotene, cream, and rosemary (dried powder; Table 10). We believe that inspection programs for foods or additives, such as ice cream, instant noodles, guar gum, and locust bean gum, have been implemented in the past; therefore, they were not included in this study.

Testing 52 finished products and raw materials, all were found to be EtO-free. The result showed that the risk of finished food and raw materials containing EtO residues in Taiwan is low.

### 3.3. Verification of the Results

To verify the effectiveness and representativeness of K-means clustering sampling during the same research period, eight food science experts were invited to provide a suggested sampling list, and a total of 45 samples were collected, including cottage cheese, jam, sauces, and cheese. No EtO residue was detected, which suggested that K-means clustering has the same sampling effect as purposive sampling. Using K-means clustering could avoid the expenses arising from large-scale sampling due to “random sampling that requires a sufficient number of samples” or “purposeful sampling where it is difficult to determine the number of samples”. However, it can focus on high risks more scientifically and efficiently.

## 4. Discussion

### 4.1. Retaining the Border Anti-Blocking Policy

Since none of the products and raw materials evaluated in this study contained EtO residue risk, the risk of food with EtO residues in the market in Taiwan seemed to be relatively small. However, since the risk of EtO in food circulation is internationally still high, an EtO prevention policy focusing on borders should be a key to maintaining food safety at this stage.

Based on the international risk notification of EtO in food, the border of Taiwan started the “inspection of EtO in food” in August 2021 and gradually increased the sampling inspection rate for specific categories, namely, finished products or raw materials with EtO risk, as shown in Table 11. From 31 August 2021 to 30 November 2023, 4146 batches were inspected and 80 batches with EtO residues were found to be unqualified, with an unqualified rate of 1.9%. Unqualified food items included locust bean gum, instant noodles, ice cream, spices, curry products, dairy products, and other pastes and flavors. At present, the Taiwan border has strictly inspected and controlled the risky food items that have been notified internationally to achieve the purpose of comprehensively preventing the entry of problematic food into the country, thereby safeguarding the health of the people. These policies should be continued.

### 4.2. Use of K-Means Clustering to Select High-Risk Samples Can Reduce the Cost of Traditional Sampling

In the early stages of this study, to ensure domestic food safety, we reviewed food information from other countries that had reported EtO residues in the past, analyzed this information using K-means clustering, and selected 52 food samples with higher risks in sampling inspection (37 manufacturers). The method reduced the labor costs required for traditional random sampling or purposive sampling of at least 100 samples. When international warning cases of food with EtO residues appear, TFDA border control measures are immediately implemented to set up fences to block the food products or increase the sampling rate so that problematic products are immediately blocked from overseas and are not allowed to be imported. This study used K-means clustering to conduct post-market food sampling to detect EtO residues in food. The results showed that all samples qualified, hence indicating that food in Taiwan is safe.

### 4.3. Inspection May Be Planned for the Uninspected Manufacturers of Cluster 4

In this study, 52 finished products and raw materials from 37 manufacturers were sampled to confirm the presence of any EtO residue. In the previous section, we concluded that cluster 4 has the highest risk; further, the smaller the distance between a point and the cluster center, the greater the risk similarity. In this study, 212 manufacturers were in cluster 4 (Figure 4). The ED of the manufacturers in cluster 4 was approximately between 0.810 and 3.983. According to the screening principle described in Section 4.2, 52 manufacturers with the shortest ED of 0.810 were selected as priority sampling inspection objects. Sampling inspections were not conducted since some manufacturers’ products or raw materials were sold out or had insufficient inventory. In the future, with increased financial resources allocated toward conducting larger-scale sampling, it will be possible to sample and inspect manufacturers and products in cluster 4 that have not yet been assessed.

### 4.4. Other Algorithms May Be Used in the Future to Evaluate and Compare the Different Methods

This study used partitional clustering, a grouping algorithm in unsupervised learning, to solve the problem of the inability to mark data in advance because of the lack of unqualified food (including EtO) in Taiwan. K-means clustering is a partial clustering method. In addition to K-means clustering, agglomerative hierarchical clustering could also be used. It considers each sample as a cluster and continuously fuses similar samples from the bottom of the tree structure. If the number of groups generated is greater than the expected number of groups, the two groups with the closest distance would be aggregated until the number of groups falls within the conditional range. The completed clusters are presented in a tree structure called a “dendrogram” [29,30,31]. In addition, clustering algorithms are based on probability distributions, such as Gaussian mixture models. It uses multiple Gaussian probability density functions to quantify the distribution of features more accurately—that is, it is a statistical model that decomposes features into multiple Gaussian probability density functions [32,33].

All of the above methods can be used to plan future inspections, further compare their commonality (referring to the intersection of manufacturer lists under different methods), and list them as priority inspection objects. Artificial intelligence and machine learning can support the implementation of inspection management, helping ensure adherence to standards.

### 4.5. Research Limitations

Owing to the legal requirements related to government data, the names of manufacturers and products in this study cannot be presented in this paper to ensure information security and protection of sensitive data.

Empty capsules for food use are produced by four pharmaceutical factories in Taiwan. Previously, these factories primarily adhered to international drug regulations, which set the upper limit of EtO residue at “not to exceed 1 mg/kg”. After receiving counseling and making necessary corrections, they are now in compliance with the regulations. In addition, prior to conducting this study, inspections were conducted for EtO residue levels in domestically produced ice cream, instant noodles, guar gum, locust bean gum, and xanthan gum. The results indicated that the risk of EtO residue in these domestically manufactured products was low (EtO-free). Therefore, these products and raw materials are not included in the scope of this study. However, since the international residual risk of EtO remains high, stringent border control measures will continue to be implemented in order to ensure food safety.

## 5. Conclusions

The following conclusions were made from the study:The use of international food safety alerts to strengthen border control can effectively ensure domestic food safety.The use of K-means clustering to construct a risk analysis model that would verify the results of purposive sampling to ensure the safety of the product after it is launched could serve as a reference for food control agencies when planning similar projects.

Based on the results of the 52 samples tested in this study, EtO-contaminated products from the international alert appear to be relatively uncommon in Taiwan. Further analysis of the sources of the samples revealed that some were domestically produced in Taiwan while some were from abroad. This indicated that the use of EtO as a pesticide or fungicide to treat raw materials or finished products is relatively rare in Taiwan, or that the food manufacturers selected in this study used compliant raw materials and implemented self-management as per law. The forecasting model of this study may be further adjusted by adding risk parameters or factors and adjusting algorithms to identify the target manufacturer or product item more accurately. Future studies should aim to develop methods capable of accurately detecting problematic manufacturers and products to help ensure food safety.

Industrial food production has created global fears over food safety. Although the methods for collecting risk data may vary for different hazardous substances, the principles of using big data for risk analysis have not changed. With an initial grasp of internationally contaminated products and raw materials matched with food cloud big data and cross-ministerial information, we could identify domestic manufacturers and suppliers who sell the same products and raw materials. Moreover, through the risk analysis model established in this research, high-risk manufacturers and products could be identified and included in the inspection projects of public health departments. In addition, based on the results of border and domestic inspections, the risk analysis model could be further used to screen high-risk manufacturers and products. This “two-way feedback” model could strengthen border and domestic market supervision, effectively intercepting unqualified products; check domestic food; and ensure food safety.

## Figures and Tables

**Figure 1 foods-13-02510-f001:**
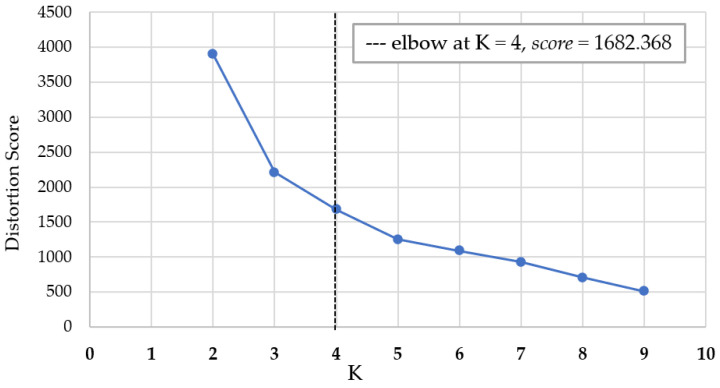
Schematic diagram of distortion score elbow for K-means clustering.

**Figure 2 foods-13-02510-f002:**
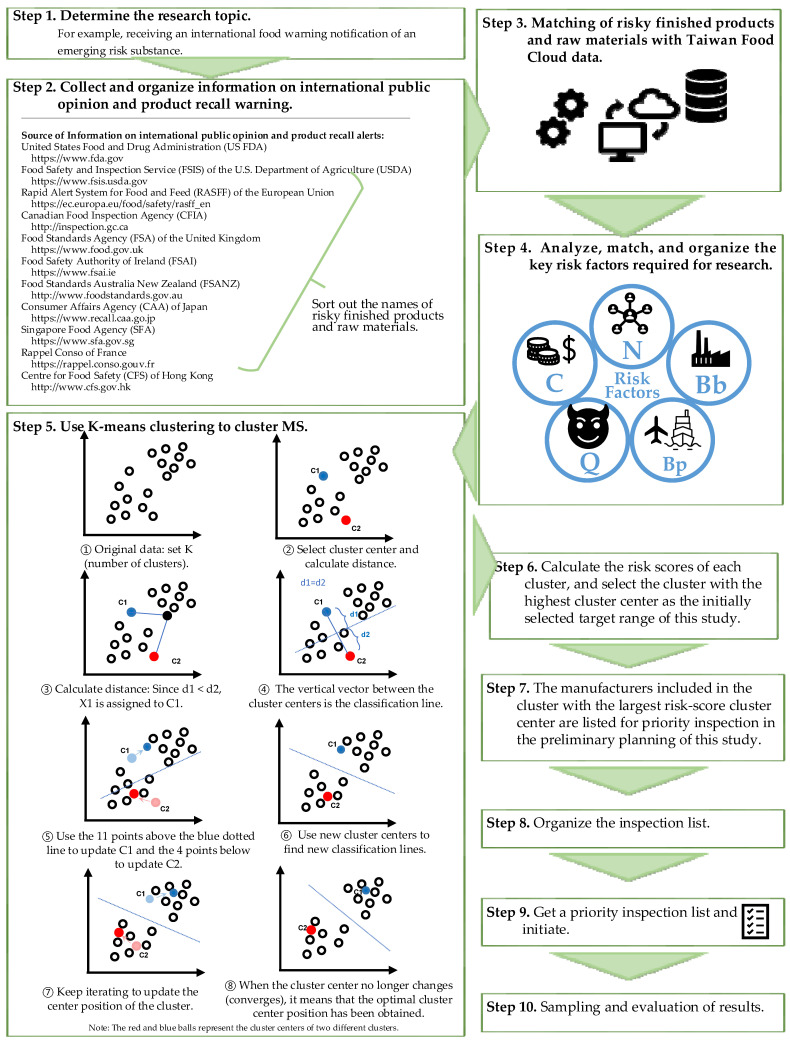
Flow chart of this study.

**Figure 3 foods-13-02510-f003:**
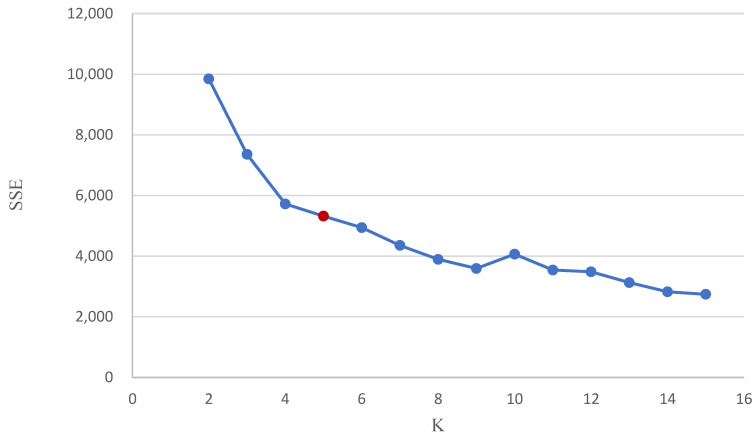
Diagram of distortion score elbow for K-means clustering. (The red circle represents K = 5).

**Figure 4 foods-13-02510-f004:**
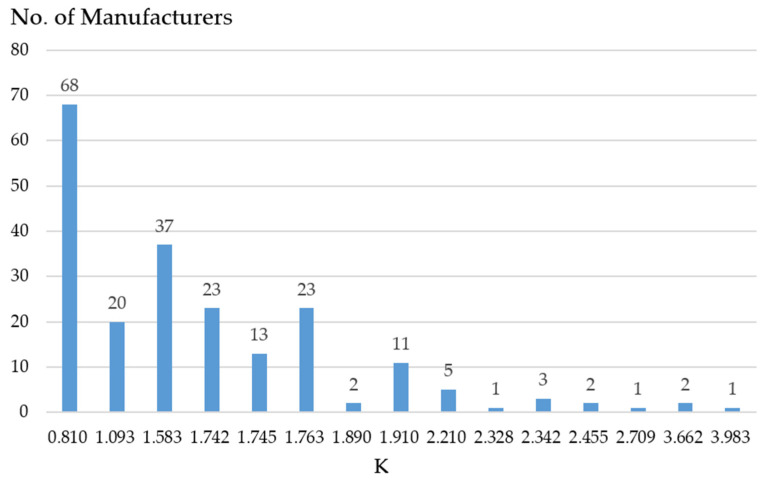
Euclidean distance distribution of manufacturers in cluster 4.

**Table 1 foods-13-02510-t001:** List of non-compliant samples of previous post-market inspection.

No.	Name of Category	Sample Name	Origin	EtO Value (mg/kg)
1	Sesame raw materials	Black sesame (Premium product)	India	1.340
2	Spicy seasoning	Ban Cabe Chili Powder—extra spicy level 30	India	14.610
3		McCormick Garlic Mixed Spice Seasoning Powder	USA	0.100
4		BonCabe Seasoned Chili Powder—medium spicy level 10	Indonesia	39.340
5		Xianguo brand Indian curry powder	Nine countries *	16.497
6	Cottage cheese	Formaggio Fresh Mozzarella	USA	1.407
7		Sonoma Sincerely Brigitte Variety Cheese	USA	0.200
8	Instant noodles (noodles, oil packets, or powder packets)	Penang Ah Lai White Curry Noodle—noodle/sauce packets	Malaysia	0.065/0.084
9		Indomie Instant Noodles Rasa Ayam Special (Special Chicken Flavor)—seasoning powder packet	Indonesia	0.187

Note: * Products manufactured in Taiwan. The factory inspection results showed no EtO risk in the manufacturing process, and the cause of this problem is likely from the raw material source countries.

**Table 2 foods-13-02510-t002:** The five main systems of Taiwan Food Cloud.

Data Source	Supplementary Note
Registration Platform of Food Business System (RPFBS)	According to Article 8 of the Taiwan Food Safety and Sanitation Management Act, food businesses whose category and scale have been announced need to register with RPFBS to operate. The system records the basic information of food companies in different industries, such as the product category and name with the largest turnover, business category, etc.
Food Traceability Management System (FTMS)	According to Article 9 of the Taiwan Food Safety and Hygiene Management Law, food businesses whose categories and scales have been announced must electronically declare and establish the source and flow of raw materials, semi-finished products, and finished products. Once a food safety incident occurs, the food industry and health authorities can quickly deal with the problematic products. This study used the raw materials and finished products declared by FTMS to connect risky raw materials and the finished products that may be related to international public opinion to obtain a list of operators that may have EtO risks.
Inspection Management System (IMS)	The database of this system integrates product test data from central health authorities and local government health bureaus.
Product Management Decision System (PMDS)	The database of this system is the integrated data established by the central health authority and the local government health bureau after the inspection and testing of food in the post-market stage. The content of the information includes the name of the product, the sampling site, the country of origin, and the inspection and test results.
Import Food Information System (IFIS)	TFDA established this system for the inspection of imported food in 2011. The database content of the system includes imported food inspection information, such as product name, country of origin, net weight, inspection method, and test results.

**Table 3 foods-13-02510-t003:** Finished products or raw materials with EtO residues in the international risk notification (data collection period: 1 January 2021 to 31 March 2023).

Serial Number	Finished Products or Raw Materials with EtO Residues		Submitted Number of Notifications	Total Number of Notifications
1	Sesame and its products	Sesame	461	470
2		Sesame paste	5	
3		Sesame oil	4	
4	Locust bean gum		57	57
5	Xanthan gum		21	21
6	Guar gum		12	12
7	Baked goods or ingredients	Bread	13	21
8		Pastry	3	
9		Biscuit	2	
10		Baking powder	1	
11		Other baked cooking and steaming foods	2	
12	Ice cream		21	21
13	Instant noodles		31	31
14	Vanilla		20	20
15	Ginger (powder)		13	13
16	Turmeric		13	13
17	Curry products	Curry powder	5	10
18		Curry	4	
19		Curry paste	1	
20	Chili		9	9
21	Black pepper		9	9
22	Psyllium		7	7
23	Cinnamon		5	5
24	Fenugreek		4	4
25	Dairy and its products	Milk powder	2	5
26		Cheese	1	
27		Fermented milk	1	
28		Cream	1	
29	Noodle products and their raw materials	Noodle	2	4
30		Flour	2	
31	Capsule		4	4
32	Calcium carbonate		7	7
33	Sodium carboxymethyl cellulose		2	2
34	Barbecue products	Barbecue powder	1	3
35		Barbecue sauce	2	
36	Ketchup		2	2
37	Ashwagandha		4	4
38	Okra		4	4
39	Amaranth		3	3
40	Onion		3	3
41	Dehydrated vegetables		5	5
42	Moringa		5	5
43	Chia seeds		2	2
44	Chickpeas		2	2
45	Barley grass		2	2
46	Kidney bean		2	2
47	Shallot		2	2
48	Gotu Kola		2	2
49	Curry leaves		2	2
50	*Bacopa monnieri*		2	2
51	Coriander seeds (Caraway seeds)		2	2
52	Carob powder (Chocolate substitute)		2	2
53	Coconut products	Coconut milk	1	2
54		Coconut juice	1	
55–90	Carotene, Chocolate, Coffee beans, Buckwheat, Oat, Rosemary, Stevia leaves, Barley grass powder, Basil, Cardamom, Celery, Clove, *Coleus forskohlii*, Cranberry, Cumin seeds, Flaxseed, Ginseng, Gluten, Green beans, Hawthorn, Indian fennel, Lemon, Lemon Leaf, Lentils, Meat, Meatballs, Melatonin, Mustard seed, Nutmeg, Quinoa, Spirulina Powder, Sunflower seeds, Tangerine, Tea, Vitamin B12, Wheatgrass powder		One case of each item	

Note: 1. This table is sorted by the number of notifications from large to small. 2. A total of 81 cases had unclear names of finished products or raw materials and could not be included in the table.

**Table 4 foods-13-02510-t004:** Interpretation table of factor quantification.

Fraction	Risk Factors from Manufacturers			Risk Factors from Finished Food Products or Raw Materials	
	C	N	Bp	Q	Bb
10	More than 100 million	5	10	69–11695	10
8	30 million to 100 million	4		20–68	
6	200,000 to 30 million	3		7–19	
4	1–200,000	2		3–6	
2	0 or null	1		1–2	
1			1		1
0	-	-		-	

**Table 5 foods-13-02510-t005:** Sum of the squared errors (SSE) for the value of K.

K	Distance Score	K	Distance Score	K	Distance Score
		6	4939.79	11	3541.50
2	9842.59	7	4352.80	12	3483.60
3	7355.31	8	3892.47	13	3128.03
4	5719.75	9	3592.26	14	2823.35
5	5319.43	10	4064.35	15	2743.12

**Table 6 foods-13-02510-t006:** Iterative process.

Iterative Calculations	Cluster Center Changes				
	1	2	3	4	5
1	3.113	4.411	3.023	3.577	2.701
2	0.585	0.770	0.228	0.333	0.157
3	0.154	0.156	0.013	0.000	0.000
4	0.000	0.000	0.000	0.000	0.000

Note: Convergence is reached when there are no or minor changes in the cluster centers. The absolute coordinate changes were capped at 0.000 for all centers. The number of iterations was four. The minimum distance between the start centers was 8.307.

**Table 7 foods-13-02510-t007:** Distance between final cluster centers.

Cluster	1	2	3	4	5
1		4.589	4.168	8.162	6.232
2	4.589		4.926	4.643	2.965
3	4.168	4.926		5.664	4.518
4	8.162	4.643	5.664		2.731
5	6.232	2.965	4.518	2.731	

**Table 8 foods-13-02510-t008:** Analysis of variance.

Variable	Cluster		Error		F	Significance
	Mean Square	Degrees of Freedom	Mean Square	Degrees of Freedom		
C	974.442	4	0.726	1356	1341.361	0.000
N	259.753	4	1.451	1356	178.962	0.000
Bb	13.816	4	0.210	1356	65.849	0.000
Q	2382.174	4	1.535	1356	1551.686	0.000
Bp	0.000	4	0.000	1356	.	.

**Table 9 foods-13-02510-t009:** Final cluster center.

Variable	Cluster				
	1	2	3	4	5
C	6	6	9	9	9
N	4	4	6	6	4
Bb	0	1	0	1	1
Q	3	7	4	10	8
Bp	0	0	0	0	0
Risk score for the cluster	13	18	19	26	22
Number of manufacturers in each cluster	438	342	177	212	192

Note: There is a missing value of 27. This means that 27 manufacturers could only be independent individuals and not be classified into the five clusters in the table; however, such a result does not mean that they are risk-free.

**Table 10 foods-13-02510-t010:** Summary table of testing finished products or raw materials for EtO.

Serial Number	Name of Category	Sample Name	Product Name	Country of Origin	Is EtO Detected?	Government Uniform Invoice Number *
1	Baked goods or ingredients	Cookie	Oyster omelet potato chips	Taiwan	No	56670000
2		Flour and its products	Super Bread Flour No. 1	Taiwan	No	22522000
3			Egg square cookies	Taiwan	No	66600000
4			White bread	Taiwan	No	27610000
5			Danish pineapple croissant	Taiwan	No	73250000
6			Camel Brand Purple Camel Heart Flour	Taiwan	No	28427000
7	Cocoa	Chocolate	Dark Cocoa Mass	Taiwan	No	27359000
8			Chocolate golden candy	Malaysia	No	84150000
9	Coffee beans	Coffee beans	Fandi Roasted Special Coffee Bean Manor Italian Coffee Bean, Northern Italian Style	Taiwan	No	86017000
10			Mandheling and Brazilian coffee	Taiwan	No	86570000
11	Food Additives	Carotene	β-carotene 1% CWS/M	Australia, New Zealand	No	16930000
12			β-carotene 1% powder	China	No	5013000
13		Sodium carboxymethyl cellulose	H-Sodium carboxymethyl cellulose	China	No	86017000
14			Sodium carboxymethyl cellulose	Taiwan	No	80130000
15		Xanthan gum	Xanthan gum	Taiwan	No	3028000
16			Xanthan gum 80SF	China	No	3341000
17			Xanthan gum RD	China	No	24547000
18	Milk and its products	Butter	Salted butter	New Zealand	No	31264000
19			New Zealand salted butter	New Zealand	No	15319000
20		Cheese	Cheese powder	Taiwan	No	23520000
21			High-melt cheese (processed cheddar)	Taiwan	No	80610000
22		Cream	Unicorn Cream (No Salt Added)	Australia, New Zealand	No	70443000
23		Fermented milk	Diluted fermented milk (less sugar and fibers)	Taiwan	No	12680000
24			Original yogurt	Taiwan	No	73250000
25		Milk powder	Skimmed milk powder	Australia	No	73250000
26			Whole milk powder	New Zealand	No	16930000
27		Salad sauce	Lemon yogurt salad sauce	New Zealand	No	22880000
28	Other plant products	Rosemary (dried powder)	Rosemary leaves	Taiwan	No	86384000
29			Rosemary powder	Taiwan	No	53260000
30		Stevia leaves	H-Stevia Leaf 094	Taiwan	No	86017000
31		Vanilla and its products	Vanilla chicken baking powder	Taiwan	No	73000000
32			Vanilla powder	Taiwan	No	89410000
33			Vanilla flavoring powder (collagen)	Taiwan	No	23520000
34	Healthy food (plant juices and extracts)	Healthy food with ginger and turmeric extract	Sentosa Curcumin and Zn Male Complex Tablets	Taiwan	No	12197000
35			Grape King Yi Turmeric Ex Flagship Capsules	Taiwan	No	11880000
36	Sauces	Barbecue sauce	Bill Head Barbecue Sauce (Sha Cha)	Taiwan	No	80550000
37		Ketchup	Ketchup	Taiwan	No	23930000
38			Ketchup	Taiwan	No	3028000
39			Veggie Ketchup	Taiwan	No	3028000
40			Tomato paste	China	No	24547000
41	Sesame	Sesame and its products	Natural and unsweetened Extra Thick Black Sesame Paste	Taiwan	No	81133000
42	Spices	Chili (dry powder products)	Spicy Soup Powder	Taiwan	No	24293000
43			100% Pure White Pepper	Malaysia and Indonesia	No	31264000
44			Chili powder	New Zealand	No	70470000
45		Curry ingredients	Seasoned Curry powder	Taiwan	No	86384000
46		Curry Products	Issuta Medium Spicy Instant Curry	Taiwan	No	86384000
47		Pepper	Extra hot pepper	Taiwan	No	97301000
48			Pure White Pepper AF	Taiwan	No	23930000
49	Whole grains	Buckwheat	Golden Buckwheat Tea Bags	Taiwan	No	86017000
50		Oat	Mayushan High-Fiber Large Oatmeal	Taiwan	No	81133000
51			Oats	New Zealand	No	59660000
52			Oats with red yeast rice and buckwheat	Taiwan	No	22100000

Note: * The government uniform invoice number has been de-identified.

**Table 11 foods-13-02510-t011:** EtO inspection table for foods at the border of Taiwan (data time interval: from 31 August 2021 to 30 November 2023).

Name of the Category (Start Time of Inspection)	Number of Inspection Batches	Number of Unqualified Batches	Unqualified Rate (%)
Sesame (31 August 2021)	131	0	0
Guar gum (18 October 2021)	13	0	0
Locust bean gum (18 October 2021)	17	1	5.9
Flour Instant noodles (with and without meat; 1 March 2022) Other noodles (other pasta and couscous; 1 November 2022)	1640	44	2.7
Empty capsule Empty capsules and medicinal capsules containing medicines (15 March 2022)	25	0	0
Ice cream and related products Ice cream (15 March 2022) Other edible ice (1 December 2022)	358	2	0
Spices Pepper, capsicum, all spices, and herbs (1 November 2022) Fennel, coriander seeds, turmeric, cinnamon, cardamom, and cloves (2 March 2023) Bay leaves (31 October 2023)	315	24	7.6
Curry Products (10 February 2023) Curry powder, curry cubes, and curry paste	62	1	1.6
Rice (1 April 2023) Brown rice, glutinous rice, and white rice	80	0	0
Cereals (1 April 2023) Wheat, rye, barley, maize, buckwheat, millet, and quinoa	111	0	0
Nuts (1 April 2023) Brazil nuts, cashews, almonds, hazelnuts, walnuts, chestnuts, pistachios, macadamias, ginkgo nuts, and pine nuts	75	0	0
Dried beans (1 April 2023) Peas, chickpeas, mung beans, black beans, red beans, kidney beans, lentils, and broad beans	46	0	0
Seeds (1 April 2023) Peanuts, pumpkins, sunflower seeds, and melon seeds	25	0	0
Coffee beans (1 April 2023) Roasted and unroasted	95	0	0
Dairy products (1 April 2023) Condensed milk, butter, cream, cheese, anhydrous butter, fresh milk, long-life milk, flavored milk, and yogurt	340	2	0.6
Wheat gluten powder (1 April 2023)	4	-	-
Xanthan gum (1 April 2023)	3	0	0
Chocolate products (1 August 2023) Cocoa beans, cocoa paste, cocoa butter, cocoa powder, and other chocolate preparations	122	0	0
Tomato sauce products (1 August 2023) Tomato ketchup and other tomato sauces	27	0	0
Sweetener (1 August 2023)	2	0	0
Other paste and flavors (21 August 2023)	653	6	0.9
Jam (27 September 2023)	2	0	0
Mustard oil (4 December 2023)	1	0	0
Total	4146	80	1.9

## Data Availability

The original contributions presented in the study are included in the article. Further inquiries can be directed to the corresponding author.

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
