# Peer review of "Research on Using K-Means Clustering to Explore High-Risk Products with Ethylene Oxide Residues and Their Manufacturers in Taiwan"

_foods, 2024, doi:10.3390/foods13162510_

Round 1

Reviewer 1 Report

Comments and Suggestions for Authors

In the introduction authors should specify EtO limits in Taiwan and Taiwan’s borders. In addition,  EtO has been forbidden in Europe only in the last years. Please add some information about this and the possible implication in the market (products withdrawal...)

Another aspect that can explored and discussed  is the possibile contamination of food supplements (vitamins, minerals...)  by EtO when Sodium carboxymethyl cellulose is used in capsule formulation. 

Reviewer 2 Report

Comments and Suggestions for Authors

The objective of this Manuscript (ID foods-3104389) entitled ,,Research on using K-means clustering to explore high-risk products with ethylene oxide residues and their manufacturers in Taiwan”  was to investigate whether Taiwan has similar or identical risks of EtO residues in food and to further enhance active inspection against risky products or manufacturers through big data risk analysis.

The objective of the manuscript is important from the point of view of food safety and consumer health protection. However, some major revisions can be made to improve the quality of the manuscript, which I specified below:

Page 2, Line 62-68: Those sentences belong to the conclusion section. Please, move them to the conclusion section.

The introductory part needs to be reorganized. Section 2 and 3 should be merge to the introduction section.

Material and methods should be numbered as 2.

Page 19, Lines 466-497: References are missing.

Reviewer 3 Report

Comments and Suggestions for Authors

Ethylene oxide

The paper aims to focus on the food safety issue, which, from time to time, has lost the stakeholders' focus. Decision-making introducing the probabilistic means gives governmental bodies the key to intervine in a global economy, fast to protect consumers and economically to support the food-chain suppliers and producers.

General concept comments

The article highlights food safety and decision-making. There is no weakness, the testability of the hypothesis, methodological inaccuracies, missing controls, etc.

The authors have, with high competence, accomplished the completeness of the review topic. The relevance of the review topic is considerably high, especially to developed economies, where processed food and imports comprise a high percentage of the food market. The gap in knowledge with high competence is identified. Also, the appropriateness of references is considered very accurate.

The manuscript's results are reproducible based on the details given in the methods section.

There is a proper presentation of figures, tables, and schemes by correctly showing the data.

The conclusions are consistent with the evidence and arguments presented.

Round 2

Reviewer 2 Report

Comments and Suggestions for Authors

The authors did their best to revise the manuscript according to the reviewers' comments and they have improved the quality of the manuscript.

I have no additional comments or suggestions.